# Flexible Wearable Sensors Based in Carbon Nanotubes Reinforced Poly(Ethylene Glycol) Diglycidyl Ether (PEGDGE): Analysis of Strain Sensitivity and Proof of Concept

Antonio del Bosque *, Xoan F. Sánchez-Romate *, María Sánchez and Alejandro Ureña

Materials Science and Engineering Area, Escuela Superior de Ciencias Experimentales y Tecnología, Universidad Rey Juan Carlos, C/Tulipán s/n, Móstoles, 28933 Madrid, Spain; maria.sanchez@urjc.es (M.S.); alejandro.urena@urjc.es (A.U.)
* Correspondence: antonio.delbosque@urjc.es (A.d.B.); xoan.fernandez.sanchezromate@urjc.es (X.F.S.-R.)

**Abstract:** The electromechanical capabilities of carbon nanotube (CNT) doped poly(ethylene glycol) diglycidyl ether (PEGDGE) have been explored. In this regard, the effect of both CNT content and curing conditions were analyzed. The electrical conductivity increased both with CNT content and curing temperature due to the lower gel time that leads to a lower reaggregation during curing. More specifically, the percolation threshold at 160 and 180 °C curing temperatures is below 0.01 wt.%, and this limit increases up to 0.1 wt.% at 140 °C for an 8 h curing cycle. Moreover, the strain monitoring capabilities were investigated, and the effect of contact resistance was also analyzed. The electrical contacts made with silver ink led to higher values of gauge factor (GF) but presented some issues at very high strains due to their possible detachment during testing. In every case, GF values were far above conventional metallic gauges with a very significant exponential behavior, especially at low CNT content due to a prevalence of tunneling mechanisms. Finally, a proof of concept of fingers and knee motion monitoring was carried out, showing a high sensitivity for human motion sensing.

**Keywords:** carbon nanotubes; PEGDGE; wearable sensors; electrical properties

## 1. Introduction

Nowadays, there is an increasing interest in the development of adequate inspection techniques for Structural Health Monitoring (SHM) applications. Here, sensors are used to collect data that will be processed and interpreted to create a control system throughout the life cycle of an asset. Some conventional SHM techniques such as ultrasonic, acoustic emission, guided waves, etc., are based on very complex mathematical and statistical tools and do not offer totally on-line information about the health of the structure [1–3], so it is necessary to explore other options.

Furthermore, the use of carbon nanoparticles as reinforcement in polymer matrices is increasing. The interest lies in the fact that their addition into these insulating systems allows the creation of percolating electrical networks, promoting a very prevalent enhancement of the electrical conductivity, as has been widely reported [4–6]. This significant improvement opens a way for new functionalities of these materials such as electro-magnetic interference shielding [7].

In this regard, SHM seems to be a promising application for this type of material. It is based on the effect that strain or induced damage have on the electrical properties of the system. More specifically, the electrical properties of the carbon-based nanocomposites are governed by three main effects: the intrinsic resistivity of carbon nanoparticles, the contact resistance between adjacent nanoparticles and the tunneling effect taking place between neighboring nanoparticles [8,9]. Carbon nanoparticles and especially carbon nanotubes (CNTs) show a piezoresistive behavior [10,11]. This means that their electrical resistivity changes when subjected to mechanical strain. In addition, the tunneling resistance changes in a linear-exponential way with the increasing distance between nanoparticles [12,13].

Furthermore, this prevalence of both tunneling and piezoresistive mechanisms makes the nanocomposites very sensitive to applied strain. So far, the sensing capabilities of carbon nanotube (CNT) based composites have been probed since they show a much higher Gauge Factor (GF) than other conventional strain gauges made of metallic foils as a result of their piezoresistive nature, as well as the tunneling effect [14,15]. Consequently, the use of CNT reinforced polymers as SHM systems is being extensively investigated [16].

In this regard, there is a lot of research concerning the SHM capabilities of CNT-based composites in structural applications. More specifically, the electromechanical capabilities of CNT-based nanocomposites have been widely investigated in, for example, structural epoxy matrices [17,18], where they have demonstrated good capabilities for strain sensing as both bulk materials as well as coatings [19]. However, these systems usually present very low values of failure strain. Therefore, their use as flexible sensors is often very limited and thus it is necessary to explore other options.

This work aims to investigate the sensing capabilities of flexible matrices doped with CNTs for their use in strain monitoring applications, which require much higher values of failure strain. In this context, the use of CNT or GNP-based flexible polymers and E-textiles as wearable sensors are now of interest [20–24] as they are capable of reaching high values of sensitivity for human motion monitoring. In addition, carbon nanoparticle based flexible systems have also shown promising properties as capacitive pressure sensors [25–27]. This way, this study is focused on the electromechanical analysis of poly(ethylene glycol) diglycidyl ether (PEGDGE) reinforced with CNTs. This epoxy system presents much higher values of failure strain than structural epoxy systems such as diglycidyl ether of bisphenol A (DGEBA) based ones. Furthermore, the influence of the curing conditions (both temperature and time) is also investigated in order to better understand the effects of the curing cycle on the electromechanical properties of these materials, both in terms of electrical conductivity and strain sensitivity. Finally, a proof of concept of the system for human motion monitoring is proposed to validate the materials developed for this type of application.

## 2. Experimental Procedure

### 2.1. Materials

The matrix used in this study is poly(ethylene glycol) diglycidyl ether (PEGDGE), which is an epoxy resin supplied by SigmaAldrich® (Merck, Saint Louis, MO, USA). The monomer has a viscosity of 60–110 mPa·s at room temperature with a molecular weight of 500 g/mol, therefore, it contains 9–10 ethylene oxide units approximately. The hardener is 4,4-diaminodiphenylsulfone (DDS), which is an amino-hardener with a 99% purity, supplied also by SigmaAldrich®. The monomer and hardener were used in a stoichiometric ratio of 100:25 wt.%, respectively.

Multi-Wall Carbon Nanotubes (MWCNTs) are NC7000, supplied by Nanocyl® (Sambreville, Belgium). They have an average diameter of 10 nm and a length up to 1.5 μm with a 95% purity. Nanocomposites were manufactured with CNTs embedded in the flexible epoxy resin.

### 2.2. Manufacturing of CNT/PEGDGE Nanocomposites

Twelve different CNT/PEGDGE nanocomposites have been manufactured: three materials with 0.01, 0.05 and 0.1 wt.% CNTs in combination with four different curing cycles such as 140 °C for 8 h, 160 °C for 4 h, 160 °C for 5 h, and 180 °C for 3 h. Table 1 summarizes the different conditions manufactured.

**Table 1.** Summary of nomenclature for the different nanocomposites.

| Sample Nomenclature | wt.% MWCNTs | Curing Cycle |
|---|---|---|
| 0.01 CNT-140 °C-8 h | | 140 °C for 8 h |
| 0.01 CNT-160 °C-4 h | | 160 °C for 4 h |
| 0.01 CNT-160 °C-5 h | 0.01 | 160 °C for 5 h |
| 0.01 CNT-180 °C-3 h | | 180 °C for 3 h |
| 0.05 CNT-140 °C-8 h | | 140 °C for 8 h |
| 0.05 CNT-160 °C-4 h | | 160 °C for 4 h |
| 0.05 CNT-160 °C-5 h | 0.05 | 160 °C for 5 h |
| 0.05 CNT-180 °C-3 h | | 180 °C for 3 h |
| 0.1 CNT-140 °C-8 h | | 140 °C for 8 h |
| 0.1 CNT-160 °C-4 h | | 160 °C for 4 h |
| 0.1 CNT-160 °C-5 h | 0.1 | 160 °C for 5 h |
| 0.1 CNT-180 °C-3 h | | 180 °C for 3 h |

These curing cycles were selected because in this range of temperatures, it was found that, by decreasing the curing time at 140 °C (for example, 4 and 5 h), the curing was not completed, and some parts of the mixture remained in a viscous state.

The opposite behavior was found at 160 °C and 180 °C, where increasing the time to 8 h led to a degradation of the resin. For these reasons, these combinations of temperature and time were selected, in order to ensure a proper curing and to compare between different conditions.

In the case of the CNT contents, it is well known that the highest sensitivities are achieved at contents around the percolation threshold, which was found, in this case, around 0.01 wt.%. For this reason, the contents were selected to ensure the electrical conductivity of the sensors without significant detriment to the sensitivity.

MWCNTs were dispersed in PEGDGE by ultrasonication in a Hielscher Ultrasonic Processor UP400St (Wanaque, NJ, USA) at 0.5 pulse cycles and 50% amplitude. A study of the influence of sonication time on dispersion state was carried out, taking samples at initial state, 10, 20, 30 and 40 min of sonication. After dispersion procedure, the mixtures were degasified under vacuum conditions in a magnetic mixer during 20 min at 40 °C in order to properly remove the entrapped air. Then, DDS hardener were added in 100:25 proportion and mixed at 40 °C. Finally, the mixture was placed in the corresponding metallic mold previously smeared with two layers of release agent (LOCTITE® Frekote 700NC, Henkel, Düsseldorf, Germany) and they were cured in an oven, following the four curing cycles mentioned.

*2.3. Characterization*

2.3.1. Microstructural Characterization

To evaluate the dispersion degree reached for these materials, non-cured CNT/PEGDGE mixtures were observed by Light Transmitted Optical Microscopy (TOM), without hardener (DDS) and prior curing step, in order to select the optimum conditions for the dispersion procedure. The microscope used was a Leica DMR (Wetzlar, Germany) equipped with a camera Nikon Coolpix 990 (Tokyo, Japan).

Fracture surfaces of nanocomposites at room temperature were analyzed by means of Scanning Electron Microscopy (SEM) using a S-3400N apparatus from Hitachi (Tokyo, Japan). The samples were coated with a thin layer of gold for a proper characterization.

### 2.3.2. Electrical Conductivity

DC volume conductivity was evaluated using a Source Meter Unit instrument KEITH-LEY 2410 (Cleveland, OH, USA). The electrical resistance was determined by calculating the slope of the current–voltage characteristic curve within the range of 0–100 V and three samples ($60 \times 16 \times 3$ mm$^3$) were tested per each nanocomposite. For these tests, four copper electrodes were attached to the sample surface using conductive silver ink to minimize the contact resistance (Figure 1).

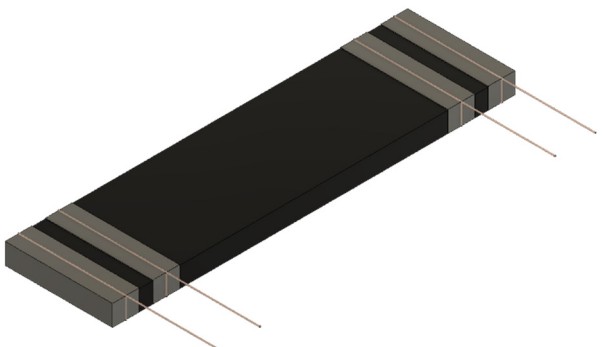

**Figure 1.** Schematics of electrode disposition in electrical conductivity measurements.

### 2.3.3. Strain Monitoring Tests

Tensile tests were conducted in a Zwick (Ulm, Germany) universal tensile machine with a load cell of 500 N. To achieve this purpose, at least five specimens of each condition were tested according to ISO 527-1:2019. The cross head speed was 10 mm/min except for the most deformable sample (140 °C for 8 h), which was tested at 30 mm/min. Strain monitoring was carried out during tensile tests by means of electrical resistance measurements between two electrodes attached to the substrate. Electrical response was recorded by using an Agilent 34410A (Santa Clara, CA, USA) module at an acquisition frequency of 10 Hz.

For the monitoring tests, two electrodes made of copper wire were attached to the substrate in two ways: (a) the conventional form, where copper wire was around the nanocomposite surface, using conductive silver ink to minimize the contact resistance, and (b) an alternative one, where copper wire was embedded in a solenoid disposition before curing. This alternative form emerged as a possible solution to the problems associated with electrode detachment at high strains that occurred with the electrodes disposition in the convectional form. Both electrode configurations are shown in Figure 2. The distance between electrodes was 30 mm.

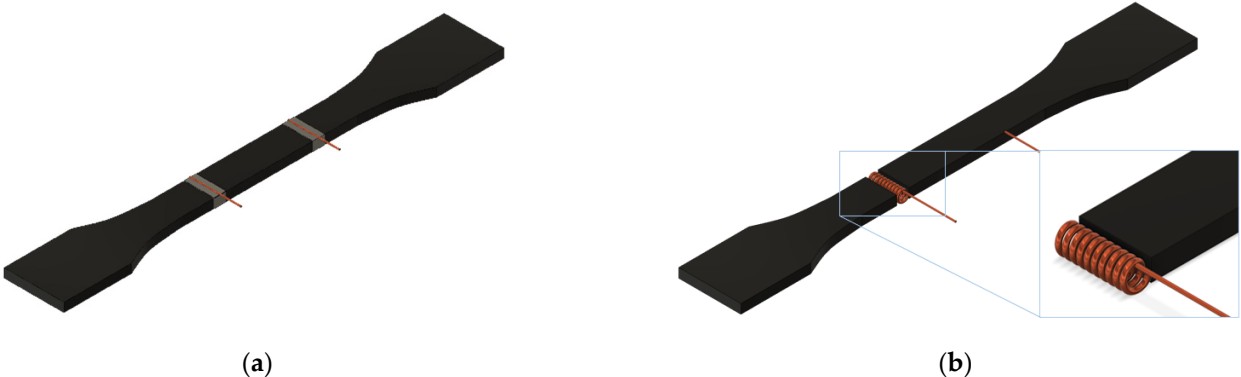

(**a**)  (**b**)

**Figure 2.** Electrode disposition in strain monitored tensile test: (**a**) conventional form (copper wire is around the nanocomposite surface with silver ink), (**b**) alternative form (copper wire is embedded in a solenoid form).

The electrical sensitivity to strain was determined by calculating the GF, defined as the change of the normalized electrical resistance ($\Delta R / R_0$), divided by the applied strain, $\varepsilon$:

$$GF = \frac{\Delta R / R_0}{\varepsilon} \tag{1}$$

Furthermore, in order to analyze the viability in biomechanical applications, tests monitoring the bending capabilities of fingers and knees were also conducted by fixing nanocomposites (dimensions of $35 \times 12 \times 1$ mm$^3$) based on CNT/PEGDGE on a nitrile glove and on a trouser leg with an adhesive layer at the ends of the gauges, respectively, as a proof of concept.

## 3. Results and Discussion

### 3.1. Microstructural Characterization and Mechanical Properties

Figure 3 shows several TOM images of the dispersion state at different sonication times for 0.01, 0.05 and 0.1 wt.% mixtures. It can be observed that, at the initial stage, the MWCNTs are mainly aggregated (left images of Figure 3). When increasing the sonication time, the agglomerates are effectively reduced, especially when comparing 10, 20 and 30 min of sonication time. However, larger sonication times (40 min) do not promote a much better CNT dispersion inside the material (right images of Figure 3). This effect can be explained by the effectiveness of the sonication technique. It has been observed in previous studies [28] that an increase in the sonication time may be detrimental to the electromechanical properties of these materials due to a prevalence of breakage mechanisms of CNT themselves, without significantly improving the CNT dispersion. In addition, the CNT content also plays an important role in the optimum dispersion conditions. In this regard, it can be elucidated that, at lower CNT contents, the agglomeration of CNTs is much more prevalent in the initial stages of the sonication process (central images of Figure 3a,b). When the sonication time is increased, there is a much more prevalent disaggregation at these lower contents due to a higher effectiveness of the sonication technique, explained by the lower viscosity of the mixture.

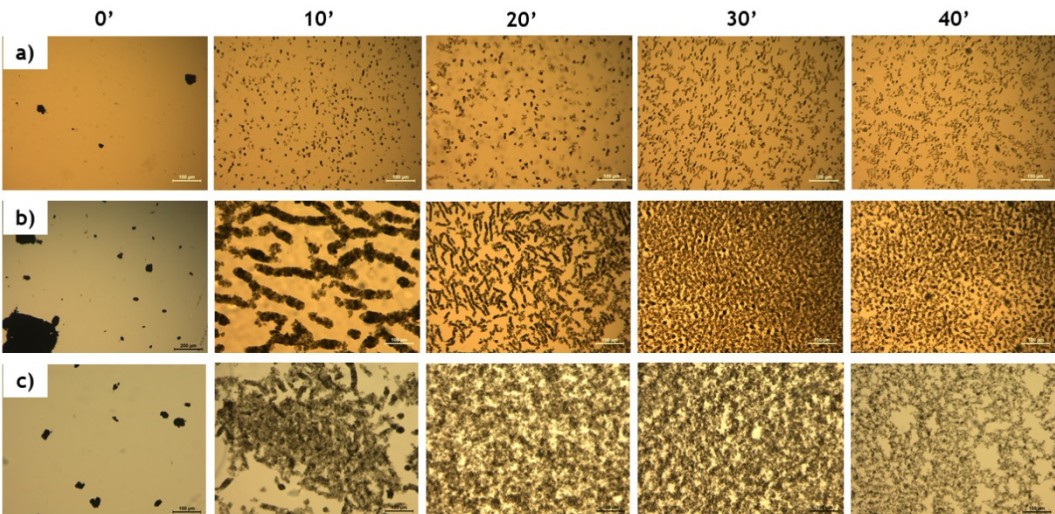

**Figure 3.** TOM images of the CNT-PEGDGE mixtures as a function of sonication time at (**a**) 0.01, (**b**) 0.05 and (**c**) 0.1 wt.% CNT content.

Therefore, the sonication time was set at 30 min as a proper CNT dispersion was observed at every condition, enough to ensure an adequate electrical network for strain sensing purposes.

Figure 4 shows the failure strain values of the nanocomposites manufactured under the different curing conditions. Here, it can be observed that the samples with 140 °C for

8 h curing show a much higher flexibility with failure strain values of around 150–200%. In this regard, the failure strain decreases from 160 °C for 4 h to 160 °C for 5 h curing due to a higher crosslinking of the PEGDGE matrix. Finally, the samples cured at 180 °C for 3 h show the lowest flexibility. However, in any case, the values of failure strain are above 20–25%, enough for their application as wearable sensors. The fracture surfaces of 140 °C for 8 h, 160 °C for 4 h, 160 °C for 5 h and 180 °C for 3 h curing are summarized in the SEM images of Figure 5. Here, a rougher fracture surface is observed in the 140 °C for 8 h and 160 °C for 4 h samples (Figure 5a,b) whereas the smoothest fracture surface is seen in the 180 °C for 3 h sample (Figure 5d), which is in good agreement with the failure strain values obtained. In addition, an increasing amount of CNTs leads to generally higher values of failure strain, which can be explained by the toughening mechanisms of the CNTs themselves [29,30].

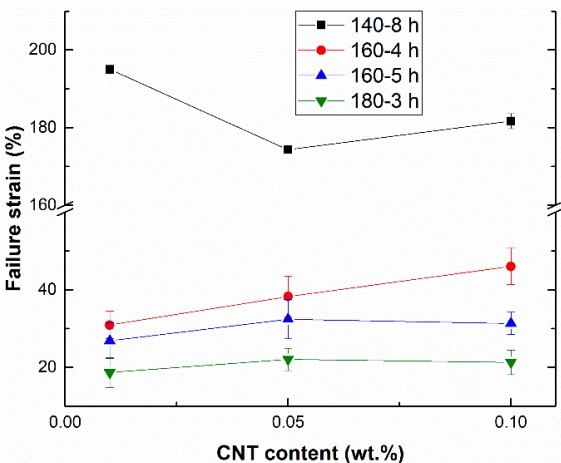

**Figure 4.** Failure strain measurements of the different tested conditions.

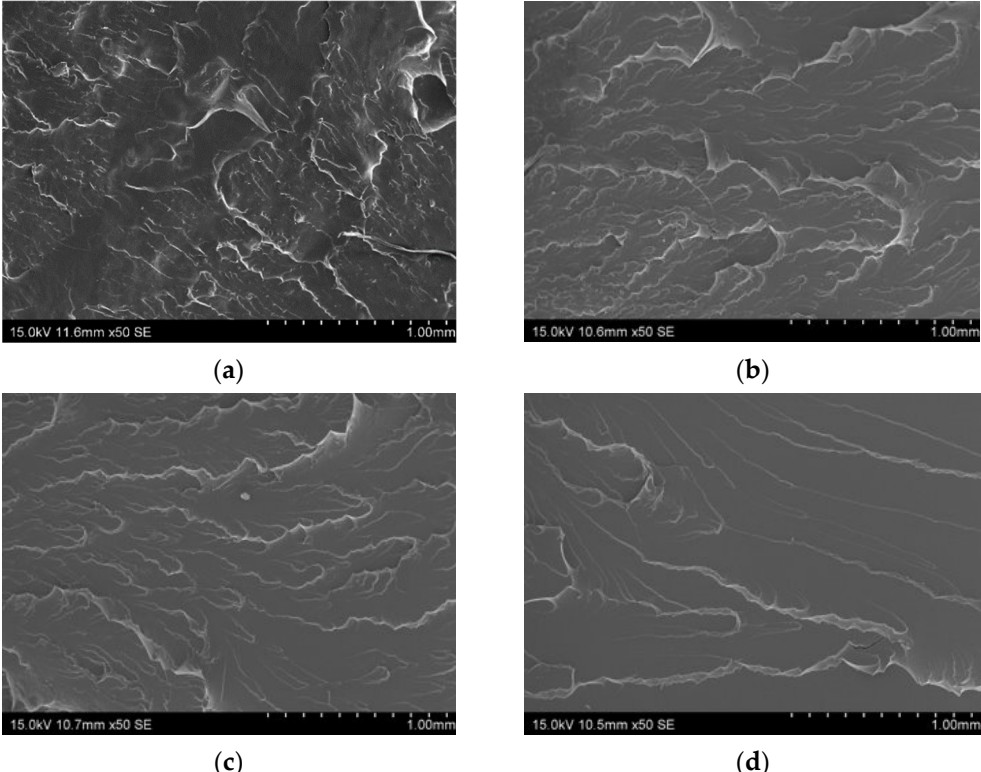

| (a) | (b) |
| (c) | (d) |

**Figure 5.** SEM images of fracture surfaces of CNT doped PEGDGE nanocomposites at (**a**) 140 °C for 8 h, (**b**) 160 °C for 4 h, (**c**) 160 °C for 5 h and (**d**) 180 °C for 3 h curing conditions.

### 3.2. Electrical Conductivity Measurements

Figure 6 summarizes the electrical conductivity values for the different tested conditions. Here, the effect of both curing cycles and CNT content can be analyzed. On one hand, it can be observed that for every CNT content, the highest values of electrical conductivity are achieved for the 180 °C for 3 h samples, whereas the lowest values of electrical conductivity are obtained in the case of the 140 °C for 8 h samples, where only the 0.1 wt.% samples were above the percolation threshold. This fact can be explained by the effect of the CNT reaggregation while curing. At higher temperatures, the gel time is lower due to the higher mobility of polymer chains. Therefore, the reaggregation of CNTs is expected to be less prevalent, as the polymer remains in a fluid state for less time. Indeed, the changing of the percolating network during curing has been observed in other studies [31,32], proving the importance of the curing parameters in the final electrical properties of the nanocomposite. In this regard, it is well known that electrical conductivity decreases when the agglomeration degree increases as there are less efficient electrical pathways inside the material, which also induces an increase of the percolation threshold, a key parameter in defining the electrical properties of the material [33]. The slight differences observed among the 4 and 5 h cured samples at 160 °C can be explained by the crosslinking degree of the polymer. To this effect, a higher crosslinking could imply the creation of more effective electrical pathways.

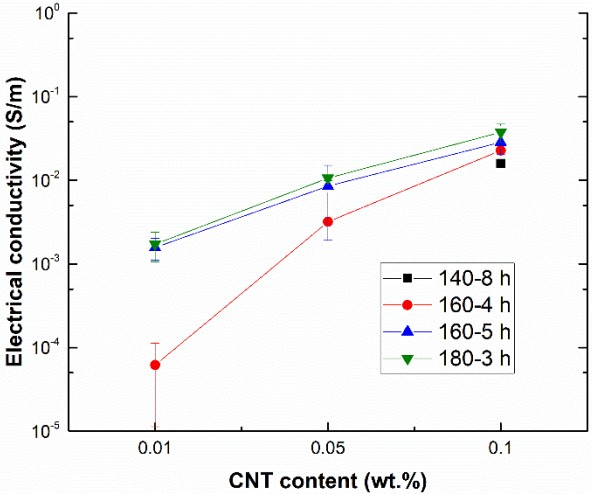

**Figure 6.** Electrical conductivity measurements for the different tested conditions.

Furthermore, as expected, an increasing amount of CNTs redounds in a higher electrical conductivity. In this context, it is important to point out that the percolation threshold for the 180 °C and 160 °C cured samples is below 0.01 wt.%, which is much lower than those previously observed in other studies using epoxy matrices with similar CNTs [18,33]. Here, the results can be attributed to the efficiency of the sonication process. More specifically, the initial viscosity of the PEGDGE resin is much lower than other epoxy systems (60–110 mPa·s to 4000 mPa·s, approximately), leading to a higher prevalence of cavitation process during sonication and inducing a drastic reduction of larger aggregates, as commented on before in the analysis of dispersion.

### 3.3. Strain Monitoring Analysis

As commented on in the Experimental Section, two types of electrode disposition were performed: embedded and silver ink attached copper wires. The aim was to understand the role of the contact resistance in the electrical sensitivity of the sensors.

In this regard, Table 2 summarizes the GF calculation for the different tested conditions depending on the electrode attachment. It can be noticed that the GF values are significantly higher when using the silver ink attached electrodes in comparison to the embedded ones.

The reason lies in the fact that the contact resistance between the copper wire and the CNT-doped substrate is much lower when using the silver ink attachment. Here, the total electrical resistance, $R_{total}$, is given by the sum of the electrical resistance of the CNT-doped material itself, $R_{nanocomposite}$, and the electrical resistance due to the contact between the electrodes and the substrate, $R_{contact}$:

$$R_{total} = R_{nanocomposite} + R_{contact} \tag{2}$$

**Table 2.** GF of the different samples with the two types of electrode's disposition.

| Sample \ Strain | GF Silver Ink Attached Wires | | | GF Embedded Wires | | |
|---|---|---|---|---|---|---|
| | 5% | 20% | 30% | 5% | 20% | 30% |
| 0.01 CNT-140 °C-8 h | | - | | | - | |
| 0.01 CNT-160 °C-4 h | 3.4 | 11.5 | 33.8 | 3.1 | 3.7 | 4.2 |
| 0.01 CNT-160 °C-5 h | 8.1 | 16.9 | - | 3.3 | 11.5 | - |
| 0.01 CNT-180 °C-3 h | 7.7 | - | - | 3.8 | - | - |
| 0.05 CNT-140 °C-8 h | | - | | | - | |
| 0.05 CNT-160 °C-4 h | 19.8 | 26.6 | 54.9 | 4.2 | 5.9 | 7.5 |
| 0.05 CNT-160 °C-5 h | 5 | 9.8 | - | 2.7 | 6.9 | - |
| 0.05 CNT-180 °C-3 h | 5 | 217 | - | 5 | 13 | - |
| 0.1 CNT-140 °C-8 h | | - | | 0.3 | 5 (100%) * | 126 (175%) * |
| 0.1 CNT-160 °C-4 h | 4.4 | 27.1 | 112.1 | 2.8 | 3.8 | 8.4 |
| 0.1 CNT-160 °C-5 h | 8.5 | 149.1 | - | 1.9 | 5.4 | 28.6 |
| 0.1 CNT-180 °C-3 h | 2 | 14.8 | - | 1.4 | 7.7 | - |

* GF for different strain levels, indicated in parentheses.

Therefore, the higher the contact resistance the lower the electrical resistance change due to the applied strain and that lowers the sensitivity.

For these reasons, silver ink seems to be an appropriate attachment for the electrodes as it reduces the contact resistance. However, the samples that were cured at 140 °C for 8 h did not show good strain-monitoring behavior when attaching the electrodes with silver ink. In this case, this behavior can be explained due to the very high failure strain of these samples. Here, the silver ink is not able to reach these strain levels, leading to an early breakage of the contacts. Therefore, the embedded technology is the best option for the monitoring of very high strains.

Some interesting facts can be found when analyzing the effect of the curing cycle and CNT content on strain sensitivity. As a general fact, the higher the curing temperature, the higher the sensitivity of the sensors. This is in good agreement with the previous statements concerning the electrical properties, as these samples showed higher values of electrical conductivity, explained by a better CNT dispersion, so there are no preferential electrical pathways throughout the agglomerates, leading to a prevalence of tunneling mechanisms and, thus, higher sensitivity. Furthermore, the samples with a 140 °C for 8 h curing cycle showed very exponential behavior with applied strain, with GF values around 0.3 at low strain levels (5%) and around 120 at very high strain levels (175%). In this case, the very high strain level leads to a much more accused exponential behavior, which explains the huge differences between the GF at low and high strain levels.

Moreover, it can be observed that an increase in the CNT amount leads to a general reduction of the sensitivity of the sensors. This is also in good agreement with other studies [9] as the distance between adjacent nanoparticles is higher for lower contents,

leading to a more prevalent electrical resistance change due to the tunneling effect. Anyway, in every case the GF values, even at lower strains, were significantly higher than those achieved for conventional metallic gauges (around 2–3), proving the great potential of the developed materials for strain monitoring purposes.

Therefore, once the strain sensing capabilities of the CNT doped PEGDGE nanocomposites had been analyzed, a proof of concept of the proposed materials for the strain monitoring of human motion was carried out.

### 3.4. Proof of Concept of Human Motion Monitoring

Figure 7 shows the electrical response under glove strain monitoring tests. In order to prove the sensitivity of the developed sensing material, two kind of finger movements were conducted. On one hand, each finger was closed and then opened (Figure 7a) and, on the other hand, the fingers were consecutively closed into a fist (Figure 7b).

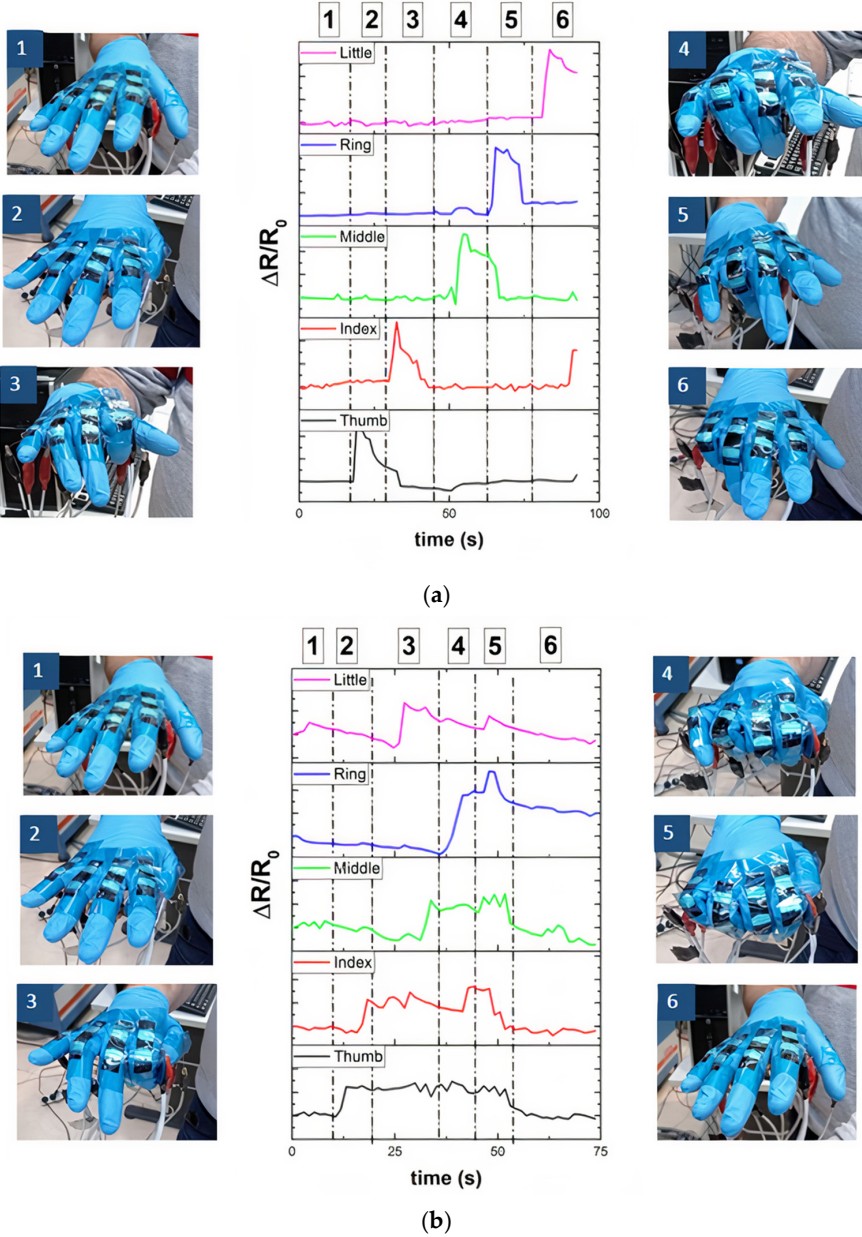

**Figure 7.** Electromechanical measurements of finger motion monitoring in the case of (**a**) fingers closing and opening and (**b**) closing fingers into a fist.

It can be observed that the closing of each finger induced a sudden increase of the electrical resistance due to the effect of the strain field over the sensor, as it was mainly subjected to a tensile strain as observed in the photos of Figure 7. Furthermore, when each finger was opened, a decrease of the electrical resistance was noticed, due to the partial recovery of the strain field in each sensor.

On the other hand, the fingers closing into a fist induced an increase of the electrical resistance that was not recovered, as expected, as the fingers remained closed. Here, it was interesting to notice that the electrical resistance of the little finger channel increased when closing the third finger (middle one, region 3 of Figure 7b). This can be explained by the effect of the closing of thumb, index and middle finger on the motion of the little one, leading to a strained field around the sensor. Finally, as expected, the resistance values were recovered when opening the hand (region 6 of Figure 7b).

In addition to the glove strain monitoring tests, knee motion monitoring was carried out. In this regard, Figure 8 summarizes the electrical response under consecutive knee bending. Here, it can be observed that the electrical resistance increased when bending the knee due to the tensile strain affecting the sensors. It can be also noticed that the electrical resistance was almost recovered when stretching the knee, indicating that the sensors were not severely damaged, as they partially recover the initial state.

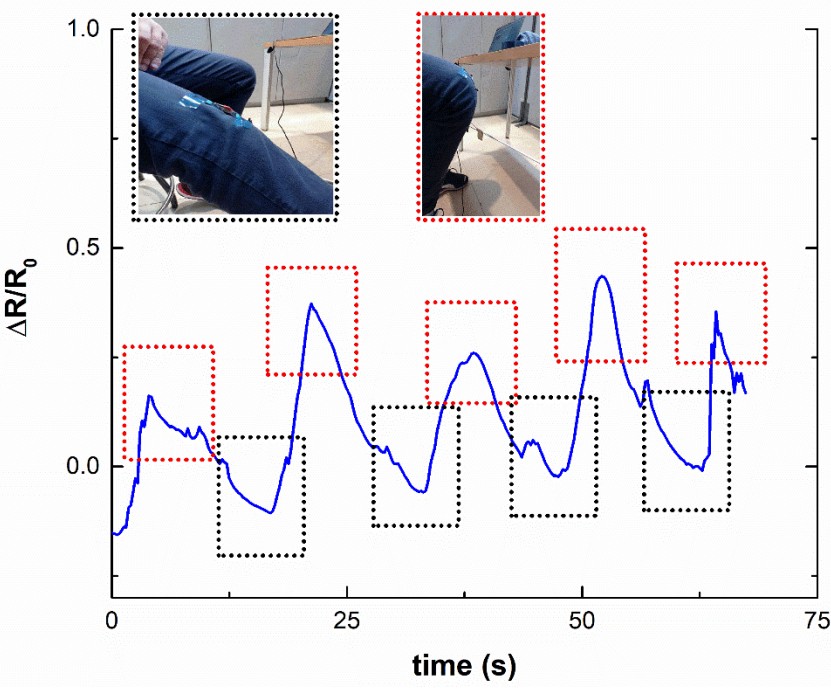

**Figure 8.** Electromechanical response of knee motion under consecutive bending cycles.

Therefore, the commented tests prove the promising applicability of the developed sensors for human motion monitoring, showing a good sensitivity together with high flexibility.

## 4. Conclusions

The mechanical and strain sensing capabilities of CNT doped PEGDGE nanocomposites have been investigated.

The optimum conditions for the sonication process were achieved at 30 min, where a good CNT dispersion was observed. This way, larger sonication times did not effectively improve this CNT distribution and may cause a prevalent breakage of CNTs themselves.

Furthermore, the effect of the curing cycle was explored. In this regard, the samples with a 180 °C for 3 h curing cycle showed the highest values of electrical conductivity but the lowest flexibility, due to a higher crosslinking of the polymer matrix. The higher

values of electrical conductivity were explained by the lower gel time in comparison to 160 and 140 °C curing cycles and, thus, there was a less prevalent CNT agglomeration during curing.

In addition, the effect of the contact resistance of the electrodes has been investigated by using embedded and silver ink attached electrodes. The silver ink disposition led to higher sensitivity, being indicative of lower contact resistance, but presented some problems of detachment at very high strain levels. In every case, the gauge factor values were above conventional metallic gauges.

Finally, a proof of concept of human motion monitoring was carried out with a proof of concept of finger and knee motion monitoring. The results proved the high sensitivity of the proposed materials, which would be very promising for strain sensing purposes.

**Author Contributions:** Conceptualization, A.d.B. and X.F.S.-R.; Formal analysis, A.d.B. and X.F.S.-R.; Funding acquisition, M.S. and A.U.; Investigation, A.d.B. and X.F.S.-R.; Methodology, A.d.B.; Writing—original draft, A.d.B. and X.F.S.-R.; Writing—review & editing, M.S. All authors have read and agreed to the published version of the manuscript.

**Funding:** This research received no external funding.

**Acknowledgments:** This work was supported by the Agencia Estatal de Investigación of Spanish Government [Project MULTIFUNC-EVs PID2019-107874RB-I00] and Comunidad de Madrid Government [Project ADITIMAT-CM (S2018/NMT-4411)].

**Conflicts of Interest:** The authors declare no conflict of interest.

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
