# Peer review of "Flexible Wearable Sensors Based in Carbon Nanotubes Reinforced Poly(Ethylene Glycol) Diglycidyl Ether (PEGDGE): Analysis of Strain Sensitivity and Proof of Concept"

_chemosensors, doi:10.3390/chemosensors9070158_

Round 1
Reviewer 1 Report
This is a novel research work relevant to the specific research community. Some simple yet important scientific findings have been outlined in the manuscript.
However, there is still a need to carry out extensive and updated literature review. Specifically results need to be analysed and explained in detail.
In Table 1, it requires explanation why such random combination of temperature and time was chosen, instead of a systemic scientific experiment where one parameter (such as time) is kept same when changing the other parameter (such as temperature) . Also please explain why the range of CNT content% was not widened.
Most of the figures need major work on presentation. Some of them can be combined for better presentation. Also figure 7 looks blurry.
Typos in line 194, 88, 228, equation 2.
Figure 5 missing SEM image at 140oC for 8 hours.
May consider looking into some relevant literatures like: https://doi.org/10.1002/adsu.202000228 https://doi.org/10.1002/9781119529538.ch2 https://doi.org/10.1021/acsnano.7b05921
Reviewer 2 Report
Present work is on the fabrication of CNT/PEGDGE compsite and its strain sensitivity measurements.
It is a nice work and should be suitable for publication in the journal.
There are several points that need to be addressed.
i) Please clarify how the curing conditions were selected. Total induced thermal energies are different, which may induce difference in the cross-linking of the polymer.
ii) For the electrical properties and GF measurements, why not use four probes to virtually eliminate the contact resistance?
It should be possible to embed four probes in the test piece as well.
iii) Please clarify how the strain gauges were positioned on the glove.
iv) 3mm strip seems too thick for wearable use as well. What happens when you make it much thinner? It may improve the cyclability of the device.
v) Fig7 doesn't seem to track the bending process of the fingers. Instead, the resistance seems to abruptly change to maximum value. Can you track the bending angle of the finger?
Reviewer 3 Report
The manuscript entitled “Flexible Wearable Sensors Based in Carbon Nanotubes Reinforced Poly(ethylene glycol) Diglycidyl Ether (PEGDGE): Analysis of Strain Sensitivity and Proof of Concept.",
reports an interesting research in which the electromechanical and deformation detection properties in nano composites, PEGDGE matrix doped with carbon nanotubes are explored.
The research was carried out with systematicity in which different samples were analyzed according to the quantities of nanotubes and the polymerization conditions.
The research topic is interesting.
The manuscript is well written and reports in a simple and clear way all the experimental phases.
The results are fully commented and discussed. Particularly interesting is the application of the results for the detection of the hands and the knee.
I have no details to send to the authors.
In light of the above, I believe that the manuscript can already be considered for its publication in the present version.
